# Paradigm Shifting of Systemic Chemotherapy for Unresectable Pancreatic Cancer in Japan

**DOI:** 10.3390/jcm8081170

**Published:** 2019-08-04

**Authors:** Junji Furuse

**Affiliations:** Department of Medical Oncology, Kyorin University Faculty of Medicine, 6-20-2, Shinkawa, Mitaka-Shi, Tokyo 181-8611, Japan; jfuruse@ks.kyorin-u.ac.jp; Tel.: +81-422-47-5511

**Keywords:** metastatic pancreatic cancer, locally advanced pancreatic cancer, hereditary pancreatic cancer, second-line chemotherapy

## Abstract

Systemic chemotherapy plays an important role in the treatment of pancreatic cancer, to improve the survival of patients with pancreatic cancer. Unresectable pancreatic cancer can be classified into three categories: metastatic, locally advanced, and hereditary pancreatic cancers. Furthermore, the second-line chemotherapy is required to prolong the survival. The combined regimens of oxaliplatin, irinotecan, fluorouracil and leucovorin (FOLFIRINOX) and gemcitabine plus nab-paclitaxel (GEM plus nab-PTX) have been recognized as the standard of care for advanced pancreatic cancer. However, the consensus of selection of the first-line chemotherapy still remains. Randomized controlled trials (RCTs) between FOLFIRINOX and GEM plus nab-PTX are ongoing for locally advanced and metastatic disease in Japan, respectively. Hereditary pancreatic cancer, especially associated with *BRCA* mutations, is responsive to platinum-containing regimens and/or poly (ADP-ribose) polymerase (PARP) inhibitors. It is becoming more important to examine the presence/absence of *BRCA* mutations to select the appropriate treatment strategy for individual patients. Although some S-1-based regimens have been investigated in the second-line treatment after GEM-based chemotherapy in Japan, no regime demonstrated survival benefit. Nanoliposomal irinotecan (nal-IRI) plus FF has been established as the standard of care in the second-line treatment in a global phase III trial (NAPOLI-1). A randomized phase II trial comparing FF plus nal-IRI with FF alone was also conducted in Japan to examine the efficacy and safety of the FF plus nal-IRI in Japanese patients.

## 1. Introduction

Pancreatic cancer is a disease with one of the worst prognoses; the 5 year survival rate of patients with pancreatic cancer is a dismal 5% to 10% [1]. Since it is difficult to diagnose pancreatic cancer at an early stage, 70–80% patients with pancreatic cancer already have unresectable disease, either locally advanced disease (Stage III) or metastatic disease (Stage IV), at diagnosis. Systemic chemotherapy is employed as the standard of care for unresectable pancreatic cancer, both locally advanced disease and metastatic disease. Chemoradiotherapy is also employed as the standard treatment for locally advanced disease.

Chemotherapy has recently been investigated for hereditary pancreatic cancer, including that associated with *BRCA* mutations, which is distinct from sporadic pancreatic cancer. Platinum-containing regimens and/or poly (ADP-ribose) polymerase (PARP) inhibitors are reported to be more effective against hereditary pancreatic cancer than against sporadic pancreatic cancer. The most promising regimen should be selected for patients with unresectable pancreatic cancer according to the stage and/or biological status of the disease, namely, locally advanced disease, metastatic disease, or hereditary pancreatic cancer [2].

Most patients who receive first-line chemotherapy have disease progression and/or unacceptable adverse events. Therefore, effective second-line chemotherapy is required to prolong the survival of patients with advanced pancreatic cancer.

## 2. Metastatic Pancreatic Cancer

Metastatic pancreatic cancer is usually treated by chemotherapy alone. Since 2011, the combined regimens of oxaliplatin, irinotecan, fluorouracil and leucovorin (FOLFIRINOX) and gemcitabine plus nab-paclitaxel (GEM plus nab-PTX) have been demonstrated to offer survival benefit over gemcitabine alone in each phase III trial [3,4]. These regimens have been established as the standard of care for metastatic pancreatic cancer.

FOLFIRINOX yielded better tumor responses and more prolonged survival as compared to GEM alone in a phase III trial; the median overall survival (OS) was 11.1 months in the FOLFIRINOX arm and 6.8 months in the GEM arm (hazard ratio (HR): 0.57; 95% confidence interval (CI): 0.45–0.73; *p* < 0.001) [3]. However, the toxicities, both hematological and non-hematological, of FOLFIRINOX were generally severe. The efficacy of FOLFIRINOX was also confirmed in a phase II trial conducted in Japanese patients, however, a high frequency of toxicities was observed, including febrile neutropenia in 22.2% of the patients [5]. To reduce these toxicities of FOLFIRINOX, trials of various modifications of FOLFIRINOX have been conducted in various countries (Table 1) [6,7,8,9]. A phase II trial of modified FOLFIRINOX, consisting of intravenous oxaliplatin 85 mg/m^2^, reduced-dose irinotecan at 150 mg/m^2^ from 180 mg/m^2^, fluorouracil infusion at 2400 mg/m^2^ over 46 h, with no bolus fluorouracil, was conducted in Japan [9]. This regimen showed comparable efficacy in terms of the response rate, progression-free survival (PFS), and OS, and was associated with reduced hematological toxicities, especially febrile neutropenia, which was found to occur at a lower rate of 8.7% in the Japanese patients [9]. Thus, modified FOLFIRINOX is currently considered a feasible regimen in Japanese patients.

S-1, which is an oral fluoropyrimidine, is also used for the treatment of unresectable pancreatic cancer in Japan, because a phase III trial comparing GEM alone, S-1 alone, and GEM plus S-1 demonstrated the non-inferiority of S-1 to GEM in terms of offering improved OS [11]. A modified FOLFIRINOX regimen containing S-1 in place of fluorouracil and leucovorin (S-IROX) was investigated in a phase I trial in Japan [10]. S-IROX was demonstrated to exhibit manageable toxicities and promising antitumor activity. Furthermore, S-IROX has another advantage that it does not require placement of a central venous port. Thus, S-IROX could also be an alternative treatment option for metastatic pancreatic cancer.

GEM plus nab-PTX is also employed as another standard of care for metastatic pancreatic cancer, based on a phase III trial in which it was compared with GEM alone; the median OS was 8.5 months in the GEM plus nab-PTX arm and 6.7 months in the GEM arm (HR: 0.72; 95% CI: 0.62–0.83; *p* < 0.001) [4]. While the toxicities were generally well-tolerated, grade 3/4 neutropenia, fatigue, and peripheral neuropathy were observed more frequently in patients treated with GEM plus nab-PTX than in those who received GEM alone. Since the phase III trial of GEM plus nab-PTX did not include Japanese patients, a Japanese phase I/II trial of GEM plus nab-PTX using the same starting doses as the aforementioned phase III trial was conducted to evaluate the efficacy and safety. It showed that the safety was comparable to the phase III trial [12]. Based on results of two trials of GEM plus nab-PTX, it was approved in Japan in 2014.

On the basis of these trial results, which suggest a superior risk-benefit balance of GEM plus nab-PTX (Table 2), this regimen is now widely used. However, phase III trials have revealed that the HR of FOLFIRINOX to GEM alone is better than that of GEM plus nab-PTX to GEM alone in phase III trials. It would be of value to examine which of the two regimens should be used as the regimen of first choice for metastatic pancreatic cancer, because modified FOLFIRINOX is associated with reduced toxicities. A phase II/III trial (JCOG1611) comparing modified FOLFIRINOX and S-IROX with GEM plus nab-PTX is under way in Japan (jRCTs031190009). The primary endpoint of the phase III part is overall survival to determine the superiority of modified FOLFIRINOX or S-IROX to GEM plus nab-PTX.

## 3. Locally Advanced Pancreatic Cancer

Chemoradiotherapy and chemotherapy are recognized as the standards of care for unresectable locally advanced pancreatic cancer. To date, no consensus has been reached on which is more effective, chemoradiotherapy or chemotherapy. A related trial was the LAP-07 trial conducted in France, which consisted of two randomizations [13]. In that study, the median OS from the date of the first randomization was not significantly different between chemotherapy and chemoradiotherapy; the median OS’s were 16.5 months and 15.2 months, respectively [13]. Thus, there is a tendency for GEM-based chemotherapy to be recognized as the community standard of care for locally advanced pancreatic cancer rather than chemoradiotherapy.

The LAP-07 trial included induction chemotherapy with GEM or GEM plus erlotinib prior to chemoradiation. However, while the benefit of induction chemotherapy has been reported from retrospective analyses, it has not been confirmed in prospective comparative trials. We conducted a randomized phase II trial of chemoradiotherapy using S-1 with and without induction chemotherapy with GEM (JCOG1106) [14]. This trial demonstrated that induction with GEM followed by chemoradiotherapy was less toxic, but did not yield longer survival as compared to chemoradiotherapy alone without prior induction chemotherapy.

The efficacy of FOLFIRINOX or GEM plus nab-PTX has not been examined in patients with locally advanced pancreatic cancer, because the phase III trials conducted to compare FOLFIRINOX or GEM plus nab-PTX with GEM alone only included patients with metastatic disease. In a systematic review of 13 studies of FOLFIRINOX for locally advanced pancreatic cancer, the median overall survival ranged from 10.0 months to 32.7 months across studies, with a patient-level median overall survival of 24.2 months. Furthermore, in these reviewed studies, 91 (28%) of 325 patients underwent resection after FOLFIRINOX therapy [15]. On the other hand, in one prospective single-arm study of GEM plus nab-PTX conducted for locally advanced pancreatic cancer (LAPACT), the median PFS was 10.2 months, and 15% of the patients underwent surgical resection [16]. In order to examine which regimen would be more promising for locally advanced disease, modified FOLFIRINOX or GEM plus nab-PTX, a randomized phase II trial of the two regimens is currently being conducted by the Japan Clinical Oncology Group (JCOG) (JCOG1407) [17].

## 4. Hereditary Pancreatic Cancer

Hereditary pancreatic cancer (HPC) is broadly defined by the presence of two first-degree relatives with pancreatic cancer, and accounts for 4–10% of all cases of pancreatic cancer [18]. Although *ATM* (mutation rate: 2.4%), *BRCA1* (0–1%), *BRCA2* (8–19%), *CHEK2* (2.9%), and *PALB2* (3.1–3.7%) have been identified as the genes responsible for HPC, known germline mutations account for less than 20% of all cases of HPC [18].

In regard to chemotherapy for HPC, it was suggested that cancer patients with germline mutations in the DNA repair pathways may be highly sensitive to DNA-damaging agents such as platinum agents [19,20,21]. A total of 549 individuals with metastatic pancreatic adenocarcinoma from Johns Hopkins and M.D. Anderson hospitals in the USA were examined to assess the efficacy of platinum-based chemotherapy in patients with a family history of pancreatic, ovarian, or breast cancer as compared to those without a family history of any of these cancers. The results revealed that first-line platinum chemotherapy was statistically significantly associated with a better survival in patients with a family history of one or more of these cancers [21]. Furthermore, in the individuals receiving first-line platinum therapy, the OS increased as the number of relatives with one or more of these cancers increased [21]. Thus, platinum-based chemotherapy was shown to exert promising activity against pancreatic cancer, however, prospective studies are still awaited. Therefore, we conducted a phase II trial of gemcitabine plus oxaliplatin to confirm the efficacy and establish a standard chemotherapy for patients with familial pancreatic cancer, who are not suitable candidates for FOLFIRINOX. In this study, a total of 45 patients were enrolled, and the one-year OS rate, which was evaluated as the primary endpoint, was only 27.9% (90% CI: 17.0–41.3), which did not meet the expected threshold (44%) [22]. Thus, it is necessary to examine the relationship between gene mutations, such as *BRCA* mutations, and the treatment efficacy in greater detail.

PARP inhibitors target defective DNA repair in cancers with *BRCA* 1/2 mutations by blocking the repair of single-strand breaks, while leaving the double-strand breaks, thereby causing death of the *BRCA* 1/2-mutant cancer cells. PARP inhibitors have been demonstrated to show clinical benefits in ovarian and breast cancer patients carrying *BRCA* 1/2 mutations [23,24,25].

Veliparib is an oral PARP-1/2 inhibitor and has been tried as monotherapy or in combination with a platinum-containing regimen in patients with pancreatic cancer [26,27]. Veliparib monotherapy exhibited modest activity against pancreatic cancer with *BRCA* 1/2 mutations, with no case of confirmed response and stable disease rate of 25% [26]. On the other hand, a phase I study conducted in a limited cohort of patients with *BRCA* mutations revealed promising activity of veliparib used in combination with gemcitabine plus cisplatin, with a response rate of 77.8% and median overall survival of 23.3 months [27].

In a randomized, double-blind, phase III trial (SOLO1 trial) performed to evaluate the efficacy of olaparib as maintenance therapy in patients with newly diagnosed advanced high-grade serous or endometrioid ovarian cancer, primary peritoneal cancer, or fallopian-tube cancer with *BRCA* 1/2 mutations who had shown complete or partial clinical response to platinum-based chemotherapy, olaparib was demonstrated to prolong the progression-free survival [24]. A phase III trial of olaparib as maintenance therapy after FOLFIRNOX therapy was also conducted in pancreatic cancer patients with germline *BRCA* 1/2 mutations (POLO trial). Although the candidates were limited to patients with a germline *BRCA* 1/2 mutation, 247 (7.5%) of the 3315 patients who underwent screening, it has been recently reported that in the POLO trial, olaparib yielded prolongation of the PFS in patients who had shown “partial response” or “stable disease” in response to FOLFIRINOX; the median PFS was 7.4 months in the olaparib arm and 3.8 months in the placebo arm (HR: 0.53; 95% CI: 0.35, 0.82; *p* = 0.003). [28].

Thus, obtaining information about the family history of pancreatic/breast/ovarian cancer and examination for the presence/absence of *BRCA* mutations is becoming more important, not only in cases of breast and ovarian cancers, but also in cases of pancreatic cancer, to select the appropriate treatment strategy for individual patients.

## 5. Second-Line Chemotherapy

While first-line chemotherapy with FOLFIRINOX and GEM plus nab-PTX has improved the survival in patients with unresectable pancreatic cancer, disease progression is eventually observed in almost all patients. Therefore, to improve the survival of such patients, an effective second-line chemotherapy regimen must be established. The OFF regimen, consisting of fluorouracil, folinic acid, and oxaliplatin, was compared with best supportive care (BSC) in the CONKO-003 trial, and it was demonstrated to offer better patient survival than BSC (Table 3) [29]. Subsequently, the CONKO-003 was switched to a randomized controlled trial to compare the FF regimen (fluorouracil plus folinic acid) with the OFF regimen; results of this trial also revealed a statistically significant prolongation of the survival in patients of the OFF arm as compared to the FF arm [30]. On the other hand, in a phase III trial (the PANCREOX trial), the combined regimens of oxaliplatin, fluorouracil and leucovorin (FOLFOX-6) regimen, which is similar to the OFF regimen, was found to be statistically significantly inferior to the FF regimen (Table 3) [31]. Therefore, the combined regimen of fluorouracil, leucovorin plus oxaliplatin is not recognized as a standard second-line chemotherapy.

S-1 is available as second-line treatment after GEM-based chemotherapy in Japan, and some clinical trials in Japan have compared S-1-based combination therapy with S-1 monotherapy [32,33,34]. Disappointingly, neither oxaliplatin (the SOX regimen) nor leucovorin (the GRAPE trial) as second-line treatment after GEM-based chemotherapy prolonged the overall survival as compared to S-1 monotherapy (Table 3) [32,33]. The combination of irinotecan with S-1 (the IRIS regimen) demonstrated some survival advantage in terms of the PFS and OS in a randomized phase II trial; the HR for death was 0.75 [33]. It was concluded that further clinical studies are warranted, but no phase III trial was conducted, because irinotecan was approved for the treatment of pancreatic cancer in Japan as a component of the FOLFIRINOX regimen.

Nanoliposomal irinotecan (nal-IRI) was developed to enhance the efficacy of irinotecan. Nal-IRI comprises the free base of irinotecan encapsulated in liposome nanoparticles. The liposome is designed to keep irinotecan in the circulation—sheltered from conversion to its active metabolite (SN-38)—for longer than free (unencapsulated) irinotecan, so that higher intratumoral levels of irinotecan and SN-38 are maintained for a longer duration as compared to free irinotecan [35]. A phase III trial comparing FF plus nal-IRI with nal-IRI alone or FF alone was conducted as a global cooperative trial (NAPOLI-1). The combination of FF plus nal-IRI yielded a better PFA and OS as compared to FF alone or nal-IRI alone, and the difference in survival between the FF plus nal-IRI arm and the FF-alone arm was statistically significant (Table 3) [34]. Nal-IRI has been approved for the treatment of pancreatic cancer in various countries in the world, and it is recognized as a standard of care for second-line therapy after GEM-based chemotherapy for patients with pancreatic cancer. Although Japan did not participate in the NAPOLI-1 trial, a randomized phase II trial comparing FF plus nal-IRI with FF alone was also conducted in Japan to examine the efficacy and safety of the FF plus nal-IRI in Japanese patients [36].

## 6. Future Perspectives of Chemotherapy for Unresectable Pancreatic Cancer

Even after the development of the intensive chemotherapy regimens of FOLFIRINOX and GEM plus nab-PTX, it remains difficult to obtain cure in patients with unresectable pancreatic cancer; few survivors remained at the end of 3 years in all phase III trials of FOLFIRINOX or GEM-nab-PTX. Therefore, attempts at obtaining a cure in patients with unresectable pancreatic cancer would inevitably necessitate surgery, the so-called conversion surgery. Satoi et al. [37] reported that among patients who received the initial treatment with GEM- and/or S-1-based chemotherapy and/or chemoradiotherapy for more than 240 days, those who underwent conversion surgery showed better survival as compared to those who did not undergo conversion surgery. Suker et al. [15] examined the efficacy of FOLFIRINOX for locally advanced pancreatic cancer in a systematic review of 13 studies involving 689 patients. In this study, 91 (28%) of 325 patients underwent resection after FOLFIRINOX, although, the long-term prognosis still remains to be confirmed. Barenboim et al. [38] reported clinical and pathologic efficacy of neoadjuvant FOLFIRINOX for unresectable locally advanced and borderline resectable pancreatic cancer. Although only 3 (10%) of 30 patients with unresectable disease were converted to resectable, 20 (87%) of 23 patients with borderline resectable disease underwent resection. Neoadjuvant FOLFIRINOX achieved complete pathologic response in 13% of resected tumors, and neoadjuvant FOLFIRINOX would be expected to extend to resectable disease. Recently, Yoo et al. [39] reported clinical outcomes of conversion surgery after neoadjuvant chemotherapy with gemcitabine-based regimens and FOLFIRIOX in patients with borderline resectable and unresectable locally advanced pancreatic cancer, compared with upfront surgery. Conversion surgery after neoadjuvant chemotherapy showed a significantly lower incidence of post-operative complications than upfront surgery. As a result, it was concluded that conversion surgery is a feasible and effective therapeutic strategy. As intensive chemotherapies such as FOLFRINOX and GEM plus nab-PTX have just been established for patients with unresectable pancreatic cancer, the benefits of conversion surgery in unresectable pancreatic cancer patients still needs to be investigated in detail.

Immune checkpoint inhibitors have recently been demonstrated to prolong the survival in patients with various advanced cancers, including melanoma and non-small cell lung cancer. Two types of immune checkpoint inhibitors, namely, anti-cytotoxic T-lymphocyte-associated protein 4 (CTLA-4) antibody and anti-programmed cell death (PD)-1 or PD ligand 1 (PD-L1) antibody, have been developed and been administered as monotherapy or combination therapy. To date, although some immune checkpoint inhibitors have been investigated as monotherapy in patients with unresectable pancreatic cancer who have previously received standard chemotherapy, none of these agents have ever exerted promising anti-tumor activity against pancreatic cancer. It was reported that no response was observed in clinical trials of ipilimumab or tremelimumab, both anti-CTLA-4 antibodies (Table 4) [40,41]. Furthermore, durvalumab, an anti-PD-L2 antibody, or durvalumab plus tremelimumab also failed to exert the expected efficacy in a randomized phase II trial [42]. Furthermore, disappointingly, combined immune checkpoint inhibitor therapy with chemotherapy, for example, the combination of nivolumab with GEM plus nab-PTX, was also found to not exert any promising efficacy in a phase I trial [43].

In order to improve the efficacy of immune checkpoint inhibitors against pancreatic cancer, various strategies have been attempted, including the combination of this class of drug with intensive chemotherapy or radiotherapy. The abscopal effect is a phenomenon in which local radiotherapy is associated with the regression of metastatic cancer at a distance from the irradiated site. [44]. The abscopal effect may be mediated by activation of the immune system. Thus, clinical trials of combined immune checkpoint inhibitor therapy with radiotherapy have been conducted for patients with pancreatic cancer. On the other hand, intensive chemotherapy using platinum agents to damage DNA would produce neoantigens, which are well-known to enhance the efficacy of immune checkpoint inhibitors. A phase II trial of the combination of nivolumab, an anti-PD-1 antibody, with FOLFIRINOX is currently under way in Japan (JapicCTI-184230).

## Figures and Tables

**Table 1 jcm-08-01170-t001:** Efficacy and safety of original FOLFIRINOX or modified FOLFIRINOX regimens for metastatic pancreatic cancer.

	Original FOLFIRINOX		Modified FOLFIRINOX				S-IROX
	Conroy et al. (2011) [3]	Okusaka et al. (2014) [5]	Mahaseth et al. (2013) [6]	Stein et al. (2016) [7]	Li et al. (2017) [8]	Ozaka et al. (2018) [9]	Shiba et al. (2015) [10]
Country	France	Japan	USA	USA	China	Japan	Japan
*N*	171	36	36	37	62	69	18
Regimen							
Irinotecan (mg/m^2^)	180	180	180	135	135	150	150 or 165
Oxaliplatin (mg/m^2^)	85	85	85	85	68	85	85
bolus fluorouracil (mg/m^2^)	400	400	0	300	0	0	-
continuous infusion fluorouracil (mg/m^2^)	2400	2400	2400	2400	2400	2400	S-1: 50 or 60 mg, oral, twice daily
Objective response rate	31.6%	38.9%	25%	35.1%	32.5%	37.7%	57.1%
Disease control rate	70.2%	69.4%	-	86.5%	60%	78.3%	92.9%
Median progression-free survival	6.4 months	5.6 months	8.5 months	6.1 months	7.0 months	5.5 months	-
Median overall survival	11.1 months	10.7 months	9.0 months	10.2 months	10.3 months	11.2 months	-
Common toxicities (Grade 3/4)							
Neutropenia	45.7%	77.8%	3%	12.2%	29%	47.8%	18%
Febrile neutropenia	5.4%	22.2%	0	4.1%	0	8.7%	-
Thrombocytopenia	9.1%	11.1%	4%	9.5%	4.8%	2.9%	12%
Fatigue	23.6%	0%	13%	12.2%	13%	5.8%	-
Diarrhea	12.7%	8.3%	13%	16.2%	0	10.1%	18%
Peripheral neuropathy	9.0%	5.6%	4%	2.7%	0	5.8%	-

FOLFIRINOX, oxaliplatin, irinotecan, fluorouracil and leucovorin; S-IROX, S-1, irinotecan and oxaliplatin; PFS, Progression-free survival; OS, overall survival.

**Table 2 jcm-08-01170-t002:** Efficacy and safety gemcitabine plus nab-paclitaxel for metastatic pancreatic cancer in the Japanese phase I/II trial.

	Ueno (2016) [11]
*N*	34
Objective response rate	58.8%
Disease control rate	94.1%
Median PFS	6.5 months
Median OS	13.5 months
Common toxicities (Grade 3/4)	
Neutropenia	70.6%
Febrile neutropenia	5.9%
Thrombocytopenia	14.7%
Fatigue	-
Diarrhea	5.9%
Peripheral neuropathy	11.8%

PFS, Progression-free survival; OS, overall survival.

**Table 3 jcm-08-01170-t003:** Clinical trials of second-line chemotherapy for pancreatic cancer.

Trial	Regimen	n	Response Rate	Median PFS	Median OS	Hazard Ratio (95% CI)	*P*-value	Author (Year)
CONKO-03	Best supportive care	23	NA	NA	2.3 months	0.45	0.008	Pelzer (2011) [29]
	OFF	23	NA	NA	4.8 months	(0.24–0.83)		
CONKO-03	FF	91	NA	2.1 months	3.0 months	NA	0.014	Pelzer (2008) [30]
	OFF	77	NA	3.0 months	6.0 months			
PANCREOX	FF	54	8.5%	2.9 months	9.9 months	1.78	0.024	Gill S (2016) [31]
	FOLOX	54	13.2%	3.1 months	6.1 months	(1.08–2.93)		
-	S-1	135	11.5%	2.8 months	6.9 months	1.03	0.82	Ohkawa S (2015) [32]
	SOX	136	20.9%	3.0 months	7.4 months	(0.79–1.34)		
-	S-1	67	6.0%	1.9 months	5.8 months	0.75	0.13	Ioka T (2017) [33]
	IRIS	60	8.3%	3.5 months	6.8 months	(0.51–1.09)		
GRAPE	S-1	290	19.7%	2.8 months	7.9 months	0.98	0.756	Ioka T (2019) [34]
	TAS118	296	27.5%	3.9 months	7.6 months	(0.82–1.16)		
NAPOLI-1	Fluorouracil/folinic acid/nanoliposomal irinotecan	117	16%	3.1 months	6.1 months	0.67	0.012	Wang-Gillam A (2016) [35]
	Fluorouracil/folinic acid	119	1%	1.5 months	4.2 months	(0·49–0·92)		
	Nanoliposomal irinotecan	151	6%	2.7 months	4.9 months	-	-	

PFS, Progression-free survival; OS, overall survival; OFF, oxaliplatin/fluorouracil/folinic acid; FF, fluorouracil/folinic acid; FOLOX, fluorouracil/Leucovorin/oxaliplatin; SOX, S-1/oxaliplatin; IRIS, irinotecan/S-1; TAS118, S-1/leucovorin.

**Table 4 jcm-08-01170-t004:** Immune checkpoint inhibitors for pancreatic cancer.

	Ipilimumab [33]	Tremelimumab [34]	Durvalumab [35]	Durvalumab + Tremelimumab [35]
*N*	27	20	33	32
Response rate	0	0	0	3.1%
Disease control rate		0	6.1%	9.4%
Median PFS		1.8 months	1.5 months	1.5 months
Median OS		4 months	3.6 months	3.1 months

PFS, Progression-free survival; OS, overall survival.

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
