# Peer review of "Paradigm Shifting of Systemic Chemotherapy for Unresectable Pancreatic Cancer in Japan"

_jcm, 2019, doi:10.3390/jcm8081170_

Round 1

Reviewer 1 Report

The authors present a review of systemic chemotherapy for advanced pancreatic cancer in Japan. In the present manuscript, the authors provided an update of the results of clinical trials, with emphasis on some of the trials conducted in Japan, for different clinical stages or types of pancreatic cancer, namely metastatic pancreatic cancer (MPC), locally advanced pancreatic cancer (LAPC) and hereditary pancreatic cancer (HP).

The English writing is fine with only a few grammar or spelling suggestions.

Although the review is comprehensive, there are a few concerns that need to be addressed, before the manuscript could be considered for publication:

1.      While the authors provided quite extensive coverage of the systemic chemotherapy regimens for MP and LAPC, very limited information was given regarding the neoadjuvant chemotherapy (NACT). The authors presented a few references regarding the conversion surgery in the section of “future perspective”, but there have been quite a good amount of references regarding NACT and conversion surgery, which the authors can refer to (Changhoon Yoo, et al. Cancers (Basel) 2019, Barenboim A, et al. Eur J Surg Oncol. 2018). The authors may consider adding a section reviewing the roles of neoadjuvant chemotherapy in the treatment of pancreatic cancer.

2. Typos and other grammar suggestions:

-  Line 16-17, abstract section: “A phase III trial among GnP, 15 modified FOLFIRINOX and S-IROX using S-1 instead of fluorouracil and leucovorin (FF) is under ongoing for metastatic disease (JCOG1611) in Japan”. Please, delete “under”.

Line 108: “The first randomization was comparison between 108 GEM alone and GEM plus erlotinib as induction chemotherapy prior to chemoradiotherapy”. Please add an “a” before comparison.

- Line 253: "..., none of 252 these agents has never exerted promising anti-tumor activity against pancreatic cancer". Please change to "none of 252 these agents have ever exerted promising anti-tumor activity against pancreatic cancer

Author Response

1. While the authors provided quite extensive coverage of the systemic chemotherapy regimens for MP and LAPC, very limited information was given regarding the neoadjuvant chemotherapy (NACT).

The authors presented a few references regarding the conversion surgery in the section of “future perspective”, but there have been quite a good amount of references regarding NACT and conversion surgery, which the authors can refer to (Changhoon Yoo, et al. Cancers (Basel) 2019, Barenboim A, et al. Eur J Surg Oncol. 2018). The authors may consider adding a section reviewing the roles of neoadjuvant chemotherapy in the treatment of pancreatic cancer.

Reply

Thank you very much for the important comments. I focus on advanced pancreatic cancer, unresectable disease, in this paper, because if post-operative adjuvant treatments and neoadjuvant treatments for resectable pancreatic cancer, including border-line resectable would be included, many papers have been published, and I thought this paper would be too long.

Therefore, the title is change to “Paradigm shifting of systemic chemotherapy for unresectable pancreatic cancer in Japan.”

Regarding conversion surgery, it is treatment option for unresectable pancreatic cancer. I mention about it in this paper. I add two papers, (Yoo C, et al. Cancers (Basel) 2019, Barenboim A, et al. Eur J Surg Oncol. 2018), and discuss extension to neoadjuvant chemotherapy.

Barenboim, et al.38) reported clinical and pathologic efficacy of neoadjuvant FOLFIRINOX for unresectable locally advanced and borderline resectable pancreatic cancer. Although only 3 (10%) of 30 patients with unresectable disease were conversed to resectable, 20 (87%) of 23 patients with borderline resectable disease underwent resection. Neoadjuvant FOLFIRINOX achieved complete pathologic response in 13% of resected tumors, and neoadjuvant FOLFIRINOX would be expected to extend to resectable disease. Recenty, Yoo et al.39) reported clinical outcomes of conversion surgery after neoadjuvant chemotherapy with gemcitabine-based regimens and FOLFIRIOX in patients with borderline resectable and unresectable locally advanced pancreatic cancer, compared with upfront surgery. Conversion surgery after neoadjuvant chemotherapy showed a significantly lower incidence of post operative complocations than upfront surgery. As a result, it was concluded that conversion surgery is a feasible and effective therapeutic strategy.

2. Typos and other grammar suggestions:

- Line 16-17, abstract section: “A phase III trial among GnP, 15 modified FOLFIRINOX and S-IROX using S-1 instead of fluorouracil and leucovorin (FF) is under ongoing for metastatic disease (JCOG1611) in Japan”. Please, delete “under”.

- Line 108: “The first randomization was comparison between 108 GEM alone and GEM plus erlotinib as induction chemotherapy prior to chemoradiotherapy”. Please add an “a” before comparison.

- Line 253: "..., none of 252 these agents has never exerted promising anti-tumor activity against pancreatic cancer". Please change to "none of 252 these agents have ever exerted promising anti-tumor activity against pancreatic cancer.

Reply

Thank you for your correcting mistakes. I correct them.

Reviewer 2 Report

A brief summary This is a review article describing the current standards for chemotherapy in the management of advanced pancreatic cancer.  

Broad comments While providing a good summary of the major trials, the organization of the review is difficult to follow and would benefit from significant restructuring.  

Specific comments 

Abstract - much more detailed than needed for an abstract

Line 10-11, reference of 3 categories, suggests that these three groups are mutually exclusive

Line 11 discussed need for second line therapy before SOC first line therapy is referenced

Line 13 - there are 2 different abbreviations for nab-paclitaxel in this article (GnP and nab-PTX)

Line 15-18 - are ongoing clinical trials appropriate to discuss in abstract

Line 19 - is ?responsive (versus effective)

Lines 23-18 - unclear goal of these sentences - perhaps just reference NAPOLI-1

Introduction

Lines 37-38 - "both locally advanced and metastatic disease" is redundant

Line 38-39 - Chemoradiotherapy is a standard option (referencing as standard treatment suggests it is the only standard of care)

Line 46-47 - unclear goal of this statement - ?further trials into second line therapy are needed?

Metastatic pancreatic cancer

Table 1 - excellent table, but as you reference in the text that this includes studies in different countries, it would be helpful to have a line noting the countries of these therapies (or at least denoting which were done in Japan)

Lines 78-86 Is S-IROX a standard of care based upon a phase I trial? Perhaps this paragraph should all be discussed after gem/n-PTX.  Line 86 seems premature as there i an ongoing phase II/III trial

Line 93 need to include references to table 2.  Is this with the same dosing/schedule for the two trials?

Line 95-96 this statement suggests that there is a phase III trial comparing the two regimens, and we should be cautious about comparing across trials when the Von Hoff study was done in multiple countries and had a slightly less robust population.  Can reference multiple smaller trials showing similar response rates in real world-analyses.

Locally advanced pancreatic cancer

Line 113-114 is redundant over line 112 (please state the statistics)

Lines 124-125 - which group is this referencing for statistics (with or without induction), how did the comparison group do?

Line 126-127 this seems to be a bold statement based upon a phase II trial

Need to discuss the fact that all of these chemoRT trials were done in era before FOLFIRINOX/gem/nab-paclitaxel induction therapy

Line 128 - these have not been studied in PHASE III trials, however, evaluated in systemic reviews.

Line 135 - helpful to reference number of patients in LAPACT

Hereditary pancreatic cancer - consider moving this whole section after second line chemotherapy as it seems to suggest that this is a group separate from the previous paragraphs

Line 148 - would recommend removal of PARP inhiitors as discussed in following paragraph, but reference platinum sensitivity as based on data from ovarian/breast cancers

Line 158 - helpful to discuss that actual genetic mutations were unknown (unselected) and why they were not candidates for FOLFIRINOX

Line 168 - please note that this is in selected BRCA mutated patients

Line 174-178 consider moving up after section 166 where you reference benefit in ovarian/breast cancer

Line 178-183, consider discussing deficits of this trial - BRCA only, germline mutations only, screening of 3k+ patients to find these patients, that this was in response platinum based therapy (not just FOLFIRINOX) stability x 4 months

Second line chemotherapy

Line 193 - "which were" --> "was" remove first part of line 194.  CONKO (misspelled).  Lines 192-197 can be compacted into once sentences.  

Line 199-202 - can just state FOLFOX-6 was inferior to FF.

Line 202-203 - is OFF/FOLFOX not considered a second line option in Japan at all (they are considered options per NCCN in US)

Table 3 - due to formatting appears confusing (shoudl not have horizontal lines between comparisons in a single study.  also should name the acronym for tirals (so that OFF versus FOLFOX6 can be separated)

Line 208 - recommending referencing trial that led to use of S-1 as second-line treatment.

Line 214 - please include p values

Line 223 - typo - PFS

Future directions

 Line 235 - notate that this is in reference to locally advanced patients - metastatic patients are not candidates

Line 256 - would be helpful to reference actual ORR

Consider discussing metronomic dosing of more chemotherapies, or combinations of chemotherapies with other immunotherapeutic agents, or combination immunotherapies

Line 272 - this seems to be a bold statement as median OS is 11.1 months in Conroy et al 2011 - how do you believe that we will obtain advances to 20% 5-yr OS?

Author Response

Thank you very much for your comments. I revised according to comments

Abstract - much more detailed than needed for an abstract

Reply

I revise the abstract following.

Systemic chemotherapy plays an important role in the treatment of pancreatic cancer, to improve the survival of patients with pancreatic cancer. Unresectable pancreatic cancer can be classified into 3 categories: metastatic, locally advanced, and hereditary pancreatic cancers. Furthermore, the second-line chemotherapy is required to prolong the survival.

The combined regimens of oxaliplatin, irinotecan, fluorouracil and leucovorin (FOLFIRINOX) and gemcitabine plus nab-paclitaxel (GnP) have been recognized as the standard of care for advanced pancreatic cancer. However, the consensus of selection of the first-line chemotherapy still remains. Randomized controlled trials (RCTs) between FOLFIRINOX and GnP are ongoing for locally advanced and metastatic disease in Japan, respectively. Hereditary pancreatic cancer, especially associated with BRCA mutations, is effective to platinum-containing regimens and/or poly (ADP-ribose) polymerase (PARP) inhibitors. It is becoming more important to examine the presence/absence of BRCA mutations to select the appropriate treatment strategy for individual patients.

Although no regime demonstrated the survival benefit in RCTs in Japan, nanoliposomal irinotecan (nal-IRI) contained regimen has been established as the standard of care in the second-line treatment in a global phase III trial. A randomized phase II trial using nal-IRI was also conducted in Japan to examine the efficacy and safety in Japanese patients.

Line 10-11, reference of 3 categories, suggests that these three groups are mutually exclusive

Reply

NCCN guidelines 2019 recommend treatment strategies by stages including unresectable locally advanced and metastatic diseases. Hereditary pancreatic cancer such as BRCA, PALB are separately mentioned in the guidelines. I add the NCCN guidelines in this paper.

Line 11 discussed need for second line therapy before SOC first line therapy is referenced

Reply

Actually, second-line treatment is added to this paper, because it was included in the editor’s offer. However, second-line treatment is different issue from 3 categories of this paper, locally advanced, metastatic, hereditary. And as comments of the reviewer 2, this paper is too long. Therefore, I delete the description of second-line treatment.

Line 13 - there are 2 different abbreviations for nab-paclitaxel in this article

(GnP and nab-PTX)

Reply

I correct Gnp is deleted.

Line 15-18 - are ongoing clinical trials appropriate to discuss in abstract

Reply

The abstract is revised.

Line 19 - is ?responsive (versus effective)

Reply

I correct effective to responsive.

Lines 23-18 - unclear goal of these sentences - perhaps just reference NAPOLI-1

Reply

I delete this part.

Introduction

Lines 37-38 - "both locally advanced and metastatic disease" is redundant

Reply

I shorten both parts.

Line 38-39 - Chemoradiotherapy is a standard option (referencing as standard treatment suggests it is the only standard of care)

Reply

Chemoradiation is recognized as one of standard treatments for locally advanced disease.

Line 46-47 - unclear goal of this statement - ?further trials into second line therapy are needed?

Reply

This part is deleted.

Metastatic pancreatic cancer

Table 1 - excellent table, but as you reference in the text that this includes studies in different countries, it would be helpful to have a line noting the countries of these therapies (or at least denoting which were done in Japan)

Reply

I add a line about country in Table 1.

Lines 78-86 Is S-IROX a standard of care based upon a phase I trial? Perhaps this paragraph should all be discussed after gem/n-PTX. Line 86 seems premature as there i an ongoing phase II/III trial

Reply

S-IROX is under investigation in phase II/III trial in Japan, not standard of care. It is an alternative regimen of FOLFIRINOX. Therefore, it is mentioned before GEM/nab-PTX.

Line 93 need to include references to table 2. Is this with the same dosing/schedule for the two trials?

Reply

The starting doses are same for the two trials. I added the information.

Line 95-96 this statement suggests that there is a phase III trial comparing the two regimens, and we should be cautious about comparing across trials. when the Von Hoff study was done in multiple countries and had a slightly less robust population. Can reference multiple smaller trials showing similar response rates in real world-analyses.

Reply

Thank you for your comment. I agree with it. I mention only the safety in Japanese patients. Table 2 shows the efficacy in Japanese trial of GEM plus nab-PTX.

Locally advanced pancreatic cancer

Line 113-114 is redundant over line 112 (please state the statistics)

Lines 124-125 - which group is this referencing for statistics (with or without induction), how did the comparison group do?

Line 126-127 this seems to be a bold statement based upon a phase II trial Need to discuss the fact that all of these chemoRT trials were done in era before FOLFIRINOX/gem/nab-paclitaxel induction therapy

Reply

This part is shortened. Up-front chemoradiation is referencing, because the efficacy of induction chemotherapy of GEM has never been examined. This study was randomized phase II study, and we did not analyze statistically the comparison. However, induction chemotherapy using GEM would be less toxic, and we assumed the induction arm would be considered more promising if the point estimate of the hazard ratio of overall survival for the induction to no induction was below 1.186. However, this paper is review, and only the results are mentioned.

Line 128 - these have not been studied in PHASE III trials, however, evaluated in systemic reviews.

Reply

I refer systemic review in this part.

Line 135 - helpful to reference number of patients in LAPACT

Reply

I refer this abstract.

Hereditary pancreatic cancer - consider moving this whole section after second line chemotherapy as it seems to suggest that this is a group separate from the previous paragraphs

Reply

The second-line chemotherapy is deleted.

Line 148 - would recommend removal of PARP inhibitors as discussed in following paragraph, but reference platinum sensitivity as based on data from ovarian/breast cancers.

Reply

I removal PARP inhibitors in this part.

Line 158 - helpful to discuss that actual genetic mutations were unknown (unselected) and why they were not candidates for FOLFIRINOX

Reply

I am not sure the reason, but as I mention in this part, germline mutations in the DNA repair pathways may be highly sensitive to DNA-damaging agents such as platinum agents, not only family history.

Line 168 - please note that this is in selected BRCA mutated patients

Line 174-178 consider moving up after section 166 where you reference benefit in ovarian/breast cancer

Line 178-183, consider discussing deficits of this trial - BRCA only, germline mutations only, screening of 3k+ patients to find these patients, that this was in response platinum based therapy (not just FOLFIRINOX) stability x 4 months

Reply

I correct this part.

Second line chemotherapy

Line 193 - "which were" --> "was" remove first part of line 194. CONKO (misspelled). Lines 192-197 can be compacted into once sentences.

Line 199-202 - can just state FOLFOX-6 was inferior to FF. at all (they are considered options per NCCN in US)

Table 3 - due to formatting appears confusing (should not have horizontal lines between comparisons in a single study. also should name the acronym for trials (so that OFF versus FOLFOX6 can be separated)

Line 208 - recommending referencing trial that led to use of S-1 as second-line treatment.

Line 214 - please include p values

Line 223 - typo – PFS

Reply

I delete this part.

Future directions

Line 235 - notate that this is in reference to locally advanced patients -metastatic patients are not candidates

Reply

It includes metastatic patients.

Line 256 - would be helpful to reference actual ORR Consider discussing metronomic dosing of more chemotherapies, or combinations of chemotherapies with other immunotherapeutic agents, or combination immunotherapies.

Reply

I add responses of immunotherapy for pancreatic cancer in Table 3.

Line 272 - this seems to be a bold statement as median OS is 11.1 months in Conroy et al 2011 - how do you believe that we will obtain advances to 20% 5-yr OS?

Reply

It is just my expectation. I delete it.